# *Camellia sinensis* Chloroplast Fluoride Efflux Gene *CsABCB9* Is Involved in the Fluoride Tolerance Mechanism

**DOI:** 10.3390/ijms23147756

**Published:** 2022-07-14

**Authors:** Bingbing Luo, Min Guang, Wenjing Yun, Shitao Ding, Suna Ren, Hongjian Gao

**Affiliations:** 1Anhui Province Key Lab of Farmland Ecological Conservation and Pollution Prevention, Engineering and Technology Research Center of Intelligent Manufacture and Efficient Utilization of Green Phosphorus Fertilizer of Anhui Province, College of Resources and Environment, Anhui Agricultural University, Hefei 230036, China; 18119603532@163.com (M.G.); ywj4546@163.com (W.Y.); dst@stu.ahau.edu.cn (S.D.); rensuna19@163.com (S.R.); 2Key Laboratory of JiangHuai Arable Land Resources Protection and Eco-Restoration, Ministry of Natural Resources, College of Resources and Environment, Anhui Agricultural University, Hefei 230036, China

**Keywords:** *Camellia sinensis*, fluoride efflux genes, chloroplasts, fluoride tolerance

## Abstract

Soil is a main source of fluoride for plants. The tea plants (*Camellia sinensis*) accumulate excessive amounts of fluoride in their leaves compared to other plants, but their fluoride tolerance mechanism is poorly understood. A chloroplast fluoride efflux gene (*CsABCB9*) was newly discovered by using transcriptome analysis, cloned from *Camellia sinensis*, and its function was demonstrated in the fluoride detoxication mechanism in *Escherichia coli*/*Xenopus laevis* oocytes and *Arabidopsis thaliana*. *CsABCB9* is expressed in tea leaves upon F^−^ treatment. The growth of tea, *E. coli*, and *Arabidopsis* were inhibited by F^−^ treatment. However, growth of *CsABCB9*-overexpression in *E.* *coli* was shown to increase with lower fluoride content under F^−^ treatment compared to the control. Furthermore, chlorophyll, xanthophyll and soluble sugar contents of *CsABCB9*-overexpression in *Arabidopsis* were improved under F^−^ treatment compared to the wild type. CsABCB9 functions in fluoride transport, and the mechanism by which CsABCB9 improves fluoride resistance in tea is mainly chloroplast protection through fluoride efflux.

## 1. Introduction

Fluorine is one of the most important trace elements in the environment, occurring in the form of fluoride (F) in animals, plants, soil, air, and water [1]. Fluoride has dual effects on human health. On the one hand, fluoride is one of the essential trace elements for bones and teeth, but excessive intake can cause fluorosis [2]. Fluoride is not an essential nutrient element for plant growth and development. Studies have shown that excessive fluoride in plants can affect the activities of various metabolic to make leaves senescent or even die [3]. On the other hand, fluoride is mainly absorbed and utilized from the soil and the atmosphere [4]. Tea trees are characterized by their rich fluoride content, and mature tea leaves have much higher fluoride than typical plants such as wheat and rice content [5]. Tea is one of the world’s most popular beverages and contains bioactive components that are beneficial to health [6]. During tea extraction, 18–99% of the fluoride in the leaves may be released into the tea broth [7]. Therefore, drinking tea is an important method of fluoride intake. Tea drinking usually does not cause fluorosis and prevents periodontal disease.

Leaves are the main source of fluoride accumulation in the tea plant (*Camellia sinensis*) [8]. Photosynthesis, which takes place in the leaves, provides nutrients essential for the growth of the tea plant. Some studies found that a treatment with high concentrations of F^−^ reduced photosynthetic rate and stomatal conductance [9,10]. Fluoride can affect photosynthetic efficiency by preventing the dephosphorylation of thylakoid membrane proteins, especially the chlorophyll a/b protein complex [11]. In addition, fluoride binds to magnesium in chlorophyll, causing thylakoids distortion and thus inhibiting primary photosynthetic reactions. Moreover, fluoride in leaves of tea plants can inhibit the activity of ribulose 1, 5-diphosphate carboxylase and thereby carbon assimilation [9,12]. Therefore, the relative tolerance of tea plants to fluoride may be closely related to its accumulation in the leaves.

Recently, fluoride-specific channels involved in fluoride tolerance have been reported as Fluc (formerly crcB) in bacteria [13] and FEX in eukaryotes [14]. FEX proteins (including FEX1 and FEX2) constitutively expressed in *Saccharomyces cerevisiae* membranes transport fluoride more specifically than the chloride, and FEX proteins have the ability to export fluoride ions [15]. *FEX* and *Fluc* are homologous genes in different species [14] and are involved in the detoxification of fluoride. Although sequence alignment reveals *FEX* homologous genes in nine plants [16], FEX protein functions have only been reported in *Arabidopsis* and tea plants. Zhu et al. in 2019 [17] cloned a fluoride export gene *CsFEX* from tea plants and found that it plays a role in fluoride detoxification through heterologous expression. CsFEX is primarily localized in the plasma membrane, and the expression of the corresponding gene *CsFEX* can be triggered by exogenous fluoride treatment [17]. However, how the fluoride efflux and accumulation as well operate at the molecular level remains still unclear, also which regulatory mechanisms are involved.

ATP-binding cassettes (ABCs) bind ATP and hydrolyze it to generate energy to drive transport and regulate further cellular function. *ABC* transporter family genes consist of eight subfamilies and the proteins coded for can transport many substrates. Thus, they are involved in regulating practically all physiological and biochemical processes in plants [18]. In terms of protein structure, most ABC transporters have two nucleotide-binding domains (NBDs) and two transmembrane domains (TMDs) [19]. ABC transporters mainly use NBDs to hydrolyze ATP to reverse the concentration gradient of a substrate and perform transmembrane transport through TMD [20]. As a consequence, ABC transporters play an important role in plant stress resistance [18]. Overexpression of *AtABCG36* did improve desiccation and salinity tolerance in *Arabidopsis* [21]. *AtABCC5* and *AtABCG31* are involved in stress resistance in *Arabidopsis* by regulating the opening and closing of stomata in the leaf blade [22,23]. In rice, the expression of the *ABCB* family genes is induced by abiotic stress [24]. Based on those (and other) studies, it seems clear that ABC transporters regulate plant stress resistance during growth and development. However, to our knowledge, no ABC transporter genes or proteins functionally characterized from tea plants, although they are most possibly involved in the fluoride stress resistance mechanism in tea plants.

Therefore, the objective of this study was to determine whether tea trees have an ABC transporter responsible for fluoride transport and high fluoride resistance. We found that fluoride increases the content of ABC transport proteins in tea trees. Furthermore, RNA-seq analysis led us to screen *C**sABCB9*, a gene whose expression is increased by F^−^ treatment. Furthermore, we isolated *CsABCB9* from tea leaves and transformed it into *E. coli*, *Xenopus* oocytes and *Arabidopsis*, and analyzed the changes in these transformed strains after treatment with different concentrations of F^−^. This enabled us to explore the function of CsABCB9 under fluoride stress. Consequently, it was revealed that the *CsABCB9* gene plays an essential role in fluoride resistance in tea and that in fact *CsABCB9* is a fluoride efflux transporter gene that enhances F resistance by protecting chloroplasts.

## 2. Results

### 2.1. Fluoride Content in Leaves of Different Tea Varieties

Leaves are the main source of fluoride enriched in tea. Therefore, we measured F^−^ content in leaves of different tea varieties treated with NaF for 3 days by hydroponic culture. Evidently, there were significant differences in F^−^ concentration in leaves from different tea varieties. Especially the F^−^ content in leaves of Wuniuzao was highest in comparison with that of other tea varieties (Figure 1A,B). However, when the ratio between the F^−^ concentrations of the F^−^ treated tea varieties to the control condition was calculated, the highest value was seen for Shuchazao (Figure 1C). This indicates that Shuchazao is very sensitive to F^−^ treatment possibly due to a strong fluoride absorption capacity. Therefore, we selected Shuchazao as a representative tea species. The RNA of Shuchazao was extracted following the control versus NaF treatment and was analyzed by transcriptome sequencing. Here, the *ABC* transporter gene family contains 910 labeled genes, of which the ABC transporter B subfamily represents the highest number (Appendix A).

Transcriptome and the statistical analysis of differentially expressed ABC transporter family genes revealed that the ABC transporter B subfamily had three genes with increased and 11 genes with decreased expression, for a total of 14 genes. In particular *CL667* showed a 3.89-fold increase in expression (Appendix A). In addition, a preliminary analysis of FPKM (Fragments per kilo base per million mapped reads) values in transcriptome sequencing revealed that NaF treatment of tea leaves significantly increased the expression of *CL667* (Appendix A). This suggested that *CL667* may be involved in the regulation of F^−^ absorption and transport in tea plants. At the gene prediction site (http://tpdb.shengxin.ren/index.html) (accessed on 1 June 2022) [25], *CL667* was also found to code for the ATP binding cassette, subfamily B member 9, and *CL667* was thus named *CsABCB9* (TEA013868.1). 

### 2.2. Analysis of CsABCB9 Expression Patterns in Tea Plants

We further examined the relative expression patterns of *CsABCB9* in leaves and roots under different F^−^ treatments. Clearly, the expression level of *CsABCB9* in leaves under NaF stress was higher in control (Figure 2), well in agreement with the pattern from RNA-Seq (Appendix A). However, there were no significant differences in *CsABCB9* expression in roots under the different treatments (Appendix A). Furthermore, the expression of *CsFEX*, previously reported as a fluoride transport gene in tea, was induced by the external NaF (Figure 2 and Appendix A) [17]. The expression patterns of *CsABCB9* and *CFEX* in leaves were consistent. These data suggest that *CsABCB9* expression is involved in the transport of fluoride in tea leaves.

### 2.3. Characteristics and Subcellular Localization of CsABCB9

According to the Gene Structure Display Server, the total length of the *CsABCB9* cDNA sequence is 2128 bp, of which the ORF sequence is 684 bp, encoding 227 amino acids (Appendix A), and the ORF sequence of *CsABCB9* was obtained by amplification of polymerase chain reaction (PCR) which using tea leaves cDNA as template and the primer pairs (*CsABCB9*-ORF-F/R) (Appendix A). Based on SOPMA (Self-Optimized Prediction Method with Alignment) prediction, CsABCB9 has a 35.68% alpha-helix, 16.74% extended strand, 5.73% beta-turn and 41.85% random. Appendix A shows the calculated tertiary structure of the CsABCB9 protein predicted by the SWISS-MODEL website [26] and Appendix A shows the analysis by TMHMM that CsABCB9 has two transmembrane proteins. Furthermore, SMART (Simple Modular Architecture Research Tool) [27] analysis indicated that CsABCB9 contains two conserved domains typical for the ABC transporter family, namely NBDs (nucleotide-binding domains) and TMDs (transmembrane domains) (Appendix A).

The subcellular localization software Cell-PLoc 2.0 predicted that CsABCB9 is localized to chloroplasts for experimental confirmation, two vectors placed under the control of the CaMV 35S promoter were constructed expressing CsABCB9::GFP and GFP::CsABCB9, followed by infiltration of tobacco leaf epidermal cells. Microscopic observation revealed that the N-terminal fusion GFP fluorescent protein of CsABCB9 overlapped with the red autofluorescence of chloroplasts, but the C-terminal fusion GFP fluorescent protein of CsABCB9 was not expressed (Figure 3). This may be due to the C—terminal of CsABCB9 protein folded into inside, resulting in the C-terminal fusion GFP florescent protein can’t observe fluorescence. Thus, these results indicate that CsABCB9 is localized to chloroplasts.

### 2.4. Functional Analysis of CsABCB9 Overexpression in Escherichia coil and Xenopus Oocytes

The homologous recombination method was used to obtain the correct overexpression of *CsFEX*-pET28a and *CsABCB9*-pET28a. The growth of the negative control strain-empty vector, the positive control strain-*CsFEX* was a fluoride export gene [17] and the *CsABCB9* overexpressing strain was inhibited by fluoride treatment in a dose-dependent manner (Figure 4). Furthermore, *CsFEX* and *CsABCB9* overexpressing strains showed better growth under both 50 and 100 mM F^−^ treatments compared to the negative control strain (Figure 4A). Moreover, the *CsABCB9* overexpressing strain exhibited a higher survival rate under 50 and 100 mM F^−^ treatments compared to the negative control strain, which was consistent with *CsFEX* overexpressing strain (Figure 4B). Furthermore, *Escherichia coli* strains overexpressing both *CsABCB9* and *CsFEX* contained lower F^−^ content than the negative control strain, a proof that *CsABCB9* plays a role in enhancing resistance to fluoride (Figure 4C).

To measure the fluoride transport activity of CsABCB9 in eukaryotes, the oocytes were treated with 1 mM NaF using the *Xenopus* oocytes heterologous system for identification. pH 5.5, 1 mM F^−^ treated oocytes were analyzed in parallel with the fluoride transport gene (*CsFEX*) as positive controls. As shown in Figure 4D, injection of *CsFEX* and *CsABCB9* cRNA into oocytes significantly reduced fluoride uptake compared to water-injected oocytes. These data suggest that CsABCB9 is a fluoride efflux transporter in oocytes, consistent with CsFEX.

### 2.5. Improved Fluoride Resistance in Arabidopsis Overexpressing CsABCB9

Since the transgenic system for tea trees is still immature and difficult to establish, we overexpressed *CsABCB9* in the model plant *Arabidopsis* to further explore its biological function in fluoride accumulation Overexpression of *CsABCB9* was confirmed in transgenic *Arabidopsis* (OE1, OE2, OE3), but not in WT. (Figure 5A). Chlorophyll concentration, xanthophyll, soluble sugar, F^−^ accumulation, and growth phenotypes of wild-type and all transgenic lines were checked to determine whether overexpression of *CsABCB9* in plant affects growth and increases fluoride resistance (Figure 5). The overexpression lines showed increased F^−^ levels in leaves, especially under 8 mM F^−^ treatment (Figure 5C). Leaf chlorophyll concentration in overexpression lines was significantly higher under F^−^ treatment compared to wild type (Figure 5D), and xanthophyll concentrations were also significantly increased under 8 mM F^−^ treatment compared to wild type (Figure 5E). Interestingly, soluble sugar concentrations in leaves of overexpression lines were significantly higher than wild type (Figure 5F).

## 3. Discussion

In this study, we found that Shuchazao is sensitive to fluoride (Figure 1) and the ABC transporter genes whose expression is up-regulated by F^−^ were searched for by transcriptome sequencing of Shuchazao leaves (Appendix A). Previously it has been shown that ABC transporters are clearly involved in plant stress resistance [18]. Many plants exposed to F^−^ exhibit phenotypes that indicated physiological toxicity [28,29]. It is well known that tea is an F^−^ resistant crop plant and that tea leaves possess a very strong fluoride accumulation capacity [30,31,32,33,34,35]. Nonetheless, many studies have revealed that high concentrations of F^−^ do affect the physiology and biochemistry of this F^−^ resistant plant, including a reduced photosynthesis, disruption of cellular ultrastructures, and changes in the leaf antioxidant system [9,33]. Previous work has demonstrated that FEX family transporters located in the cell membrane contribute to resistance against fluoride toxicity [14,34].

The web-based structure simulation of CsABCB9 revealed that a fragment contains a conserved ABC domain and a transmembrane domain (Appendix A). Furthermore, F^−^ treatment resulted in the highest expression of *CsABCB9* in leaves compared to roots (Figure 2 and Appendix A). These observations indicate that the expression of *CsABCB9* in tea leaves is closely related to fluoride content and that *CsABCB9* expression is indeed triggered induced by external F^−^.

To further clarify the function of *CsABCB9*, *CsABCB9* was transferred to prokaryotic *E. coli*, followed by treatment with different concentrations of F^−^. Clearly, overexpression of *CsABCB9* increased fluoride resistance and decreased fluoride level (Figure 4A–C). This function was similar to the positive control *CsFEX* reported by Zhu et al., 2019 [17]. Moreover, the expression of *CsABCB9* in eukaryotic *Xenopus* oocytes also reduced intracellular F^−^ levels (Figure 4D). Additionally, CsABCB9 was present in chloroplasts (Figure 3). This suggests that CsABCB9 protects chloroplasts mainly by efflux of F^−^, thereby enhancing the fluoride resistance in tea plants.

ABC transporters belong to the integrated membrane family and have a wide variety of substrates in plants. Previously, it has been shown that overexpression of *AtABCB25* improved the Pb/Cd resistance in *Arabidopsis* [36]. *AtABCB27* and its homolog in barley were implicated in the prevention of Al toxicity [37]. To determine the mechanism by which *CsABCB9* drives fluoride resistance, overexpression lines of the model plant *Arabidopsis* were established (Figure 5A); *CsABCB9* overexpression lines were found to be more resistant to exogenous F^−^ treatment. Accordingly, chlorophyll, xanthophyll, and soluble sugar concentrations were increased in comparison to wild type (Figure 5). In agreement with the observation of subcellular localization of CsABCB9 (Figure 3), overexpression of *CsABCB9* apparently enhanced the fluoride resistance by protecting leaf chloroplasts.

In conclusion, this is the first determination and cloning of the ABC transporter family gene *CsABCB9* for fluoride specific export in tea. The expression of *CsABCB9* in tea was found to be leaf-specific and induced by external F^−^ treatment. Furthermore, overexpression of CsABCB9 in *Escherichia coli* and *Xenopus* oocytes suppressed intracellular F^−^ accumulation and thus led to improve fluoride resistance. These results directly demonstrate that chloroplast-localized CsABCB9 acts as a fluoride efflux transporter to mitigate F^−^ toxicity, and that overexpression of *CsABCB9* in *Arabidopsis* can protect leaf photosynthesis at comparably high concentrations of external F^−^.

Taken together, our previous and actual studies infer that only a few fluoride (F) transporters are expressed in tea plants to maintain the healthy growth under no F or low F condition (Figure 6A). A high concentration of F increased its level in mesophyll cells through transport F into the cells via passive absorption and active ion channels [38]. F is subsequently transported to various intracellular organelles, increasing the expression of the genes encoding the chloroplast-localized fluoride efflux transporter CsABCB9 and plasmalemma-localized CsFEX. CsABCB9 transported F from the chloroplast to the cytosol, reducing F toxicity to the chloroplast, and then CsFEX exports F from the cytoplasmic matrix into cell wall or apoplastic space to avoid fluoride poisoning. Furthermore, excess F can also be transferred and stored in vacuoles by an unknown tonoplast-localized transporters to alleviate intracellular fluoride toxicity (Figure 6B). However, the precise molecular control mechanisms of fluoride accumulation, toxicity and detoxification, and resistance in tea are worth exploring in the future.

## 4. Materials and Methods

### 4.1. Tea Tree Growing Conditions and Sample Processing

Tea seeds of Wuniuzao, Nongkangzao, Longjing43, Anji white tea, and Shuchazao were collected at Hefei Hi-Tech Agricultural Garden in China. For use in hydroponics experiments, tea seedlings with constant growth were selected and washed with double-distilled water (ddH_2_O). Tea seedlings were cultured in the growth chamber for one month in the total basic nutrient solution to allow for full root development. Tea culture conditions were as follows: 12 h light (25 ± 2 °C)/12 h dark (20 ± 2 °C), 70 ± 10% relative humidity. The nutrient solution was refreshed every 3 days and the pH was maintained at approximately 5.0 using MES [39]. Tea seedlings of the same growth were treated for 3 days with 0 (control), 5 mM NaF. After treatment, tissue samples with the different treatments were taken and immediately frozen in liquid nitrogen and stored at −80 °C for further analysis.

### 4.2. Obtaining ABC Transporter-Related Genes by Transcriptome Data

Total RNA was extracted from tea leaves with different F^−^ treatments by using a rapid RNA isolation kit (Tiangen Biotech, Co., Ltd., Beijing, China) following the manufacturer’s instructions. Then, NanoDrop ND−1000 spectrophotometer (NanoDrop, Wilmington, DE, USA) and 2100 Bioanalyzer RNA Nano chip device (Agilent Technologies, Palo Alto, CA, USA) checked the quality, purity and integrity of RNA. Finally, the same amount of RNA was sent to BGI (Beijing Genomics institution) for transcriptome analysis and information on ABC transporter family genes in tea leaves by BGI (Beijing Genomics institution) analysis was compiled into a transcriptome database and protein databases including KEGG, Swiss-Prot, NR, and COG. Differentially expressed genes (DEGs) were counted for functional analysis according to significance q-values < 0.05 and difference time (Log_2_ Fold change >1).

### 4.3. Cloning and Isolation of CsABCB9 from Tea Tree

Based on the transcriptome sequencing data, the ORF of *CsABCB9* was amplified by PCR using primers (*CsABCB9*-ORF-F: ATCAGCAATGGCTTCCTCTG; *CsABCB9*-ORF-R: TCATTCACTGATGCATTATTG). The amplified product was then purified and cloned into the cloning vector pEASY-Blunt (TransGen Biotech, Beijing, China) and sequenced. Secondary and tertiary structure models of the CsABCB9 protein were obtained from the online software TMHMM (http://www.cbs.dtu.dk/services/TMHMM/) (accessed on 1 June 2022) and SWISS-MODEL (https://swissmodel.expasy.org/) (accessed on 1 June 2022) was used to predict the results [26]. The CsABCB9 protein domain was analyzed by the SMART website (http://smart.embl.de) (accessed on 1 June 2022) [27].

### 4.4. Gene Expression Analysis by QRT-PCR

Total RNA of tea leaves was extracted using a rapid RNA isolation kit (Tiangen Biotech, Co., Ltd., Beijing, China). Then, cDNA was synthesized using PrimerScript First Stand cDNA Synthesis Kit (TaKaRa, Dalian, China). qRT-PCR was performed according to Zhu et al., 2019 [17]. SYBR Premix Ex Taq (TaKaRa, Dalian, China) was used for relative expression analysis and data was performed by the 2^−^^ΔCT^ method. *CsGAPDH* (TEA025584.1) was used as an internal reference. All primers were designed with Primer 5.0 software and primer sequences are shown in Appendix A. The procedure of qPCR was 95 °C for 30 s, followed by 40 cycles of 95 °C for 5 s, 60 °C for 30 s.

### 4.5. F Resistance of Transgenic Escherichia coli

The ORF sequence of *CsABCB9* was obtained from the cloning vector pEASY-Blunt and ligated into a pET28a expression vector by homologous recombination *CsABCB9*-pET28a, *CsFEX*-pET28a (as positive control) [17] and pET28a empty vector (as negative control) were transformed into *Escherichia coli* ROSETTA cells, respectively. The correct insertion sequence was then identified by bacterial PCR and sequencing.

To determine the effect of *CsABCB9* overexpression on the growth of *Escherichia coli* strains, pET28a empty vector, *CsFEX* and *CsABCB9* transformed into *Escherichia coli* strains were cultured in Luria-Bertani (LB) Broth Medium until the optical density at 600 nm (OD 600) reached 1.0, and each of the 10 serial dilutions of the three strains were incubated in a solid LB medium containing different concentrations of F^−^ (0, 5, 50 and 100 mM) for 12 h and the growth status of the three strains were observed. The transgenic *E. coli* strains with OD 600 of 1 were also diluted 1:1000 in liquid LB (0, 5, 50 and 100 mM) containing different concentrations of F^−^ and cultured in a vibrating incubator at 37 °C, 220 rpm/min. OD 600 values were measured at different times (2, 4, 6, 8, 10, 12, and 24 h) using a spectrophotometer, and then cells incubated for 8 h at different F^−^ concentrations were collected to determine the F content of the three strains.

### 4.6. Subcellular Localization Analysis of CsABCB9

To confirm the subcellular localization of CsABCB9, the full-length coding sequence of *CsABCB9* was amplified by PCR using primers (Appendix A) containing XbaI/BglII restriction sites. The purified PCR product was inserted into pCAM-BIA1305.EGFP, which is initiated by the CaMV 35S promoter. After validation by sequencing, the plasmids *CsABCB9*::EGFP, EGFP::*CsABCB9* and 35S::EGFP (as control) were transformed into Agrobacterium tumefaciens EHA105 strain. Bacteria were collected by centrifugation and resuspended in a solution (pH 5.7) containing 10 mM MgCl_2_, 10 mM MES, and 200 mM Acetosyringone (AS). Next, transient transformation experiments with Agrobacterium tumefaciens were performed using *Nicotiana benthamiana* plants with 4–5 true leaves (for about 1 month). Cell suspensions with a density of 0.3 (OD 600) were infiltrated into *Nicotiana benthamiana* leaves with a needleless syringe and cultured for 2 to 3 days. Tobacco epidermal cells were observed with a confocal laser scanning microscope (LSM410; Carl Zeiss), green GFP fluorescence was attracted at 488 nm and observed through a 535 nm filter, and chloroplast spontaneous red fluorescence was attracted at 543 nm and observed through a 585 nm filter, as previously reported by Yang et al., 2020 [40].

### 4.7. Functional Analysis of CsABCB9 in Xenopus Oocytes

*CsABCB9* and *CsFEX* were constructed in the oocyte expression vector pT7Ts, linearized using XbaI, and transcribed to cRNA using the Ambion mMessage mMachine T7 kit. Oocytes were injected with 50 ng of *CsABCB9* and *CsFEX* cRNA as previously described Feng et al., 2011 [41]. Oocytes were treated with antibiotics (96 mM NaCl, 1 mM MgCl_2_, 2 mM KCL, 5 mM HEPES, 50 μg/mL gentamicin, 100 μg/mL streptomycin). They were incubated in ND96 solution with (100 μg/mL streptomycin, pH 7.4). Oocytes injected with water, *CsABCB9* cRNA, and *CsFEX* cRNA were incubated in a solution containing 1 mM F^−^ at pH 5.5 for 8 h. The oocytes were then centrifuged at 5000× *g* for 15 min and resuspended three times in ultrapure water to check the F^−^ content.

### 4.8. Overexpression and Phenotypic Analysis of CsABCB9 in Arabidopsis

To further investigate the function of CsABCB9, *CsABCB9* was amplified using primers (Appendix A) and recombined into a 35S: *CsABCB9* construct. Then, the vector was transferred into the *Colombia* ecotype of *Arabidopsis* by the flower soaking method [42]. First, transgenic *Arabidopsis* lines were screened on 1/2 Murashige and Skoog (MS) agar medium containing 60 mg/L hygromycin. Next, RT-PCR assays were performed using gene-specific primers to confirm the expression of *CsABCB9* in positive transgenic *Arabidopsis* lines (Appendix A), with the *AtACTIN* (AT3G18780) gene as an internal reference. All *Arabidopsis* plants were grown in light incubators at 22 ± 2 °C, 16 h light (220 μmol m^−2^ s^−1^)/8 h dark.

Wild-type and transgenic *Arabidopsis* seeds (WT, OE1, OE2 and OE3) were inoculated on 1/2 solid medium. After 2 days of incubation at a low temperature (4 °C), these plates were moved to a light incubator with a 16 h light (220 μmol m^−2^ s^−1^)/8 h dark cycle for 7 days. All *Arabidopsis* lines were transformed in a soil matrix containing a mixture of nutrients and vermiculite for 7 days, and then treated with different concentrations of F^−^ (0, 6 mM, 8 mM) for additional 7 days.

### 4.9. Collection and Detection of Related Physiological Index Samples

Bacteria in the *E. coli* culture were collected by centrifugation at 5000 rpm for 15 min and resuspended in ddH_2_O three times. Plant samples were collected and weighed. To each of the above bacterial and plant samples, 30 mL of ddH_2_O was added and extracted in a water bath at 100 °C for 30 min. The extraction mixture was cooled to room temperature and centrifuged at 5000× *g* for 15 min. Finally, the F^−^ content in the recovered supernatant was measured by a 9609NWP fluoride ion-selective electrode and 096019 stirrer probe, as described by Gao et al., 2014 [43].

The method of chlorophyll and xanthophyll concentration in *Arabidopsis* described by Xu et al., 2013 [44]. 0.1 g plant tissues were homogenized in 805 acetone and incubated in the dark for 6 h. After the homogenate was filtered and centrifuged, the supernatant was read at 665 nm, 649 nm and 470 nm by Spectra Max plus−384 (Molecular Device, San Jose, CA, USA). The total chlorophyll and xanthophyll amount were calculated.

The total soluble sugars were determined using anthrone method [45]. 0.1 g leaf tissues were boiled in 2.5 N HCl for 3 h and decolorize the sample with activated carbon for 30 min. Then, collect supernatant by centrifugation and add anthrone regent for forming color. The intensity of color was read at 620 nm and concentration of total soluble sugars was calculated according to the standard curve of glucose.

### 4.10. Statistical Analysis Was Performed

In this study, all data were performed using IBM SPSS Statistics version 17.0 and are presented as mean ± variance (SD). Duncan’s test of one-way analysis of variance was used for statistical analysis. Significance of differences was defined by asterisks (*p* < 0.01; *p* < 0.001) or by different letters (*p* < 0.05).

## Figures and Tables

**Figure 1 ijms-23-07756-f001:**
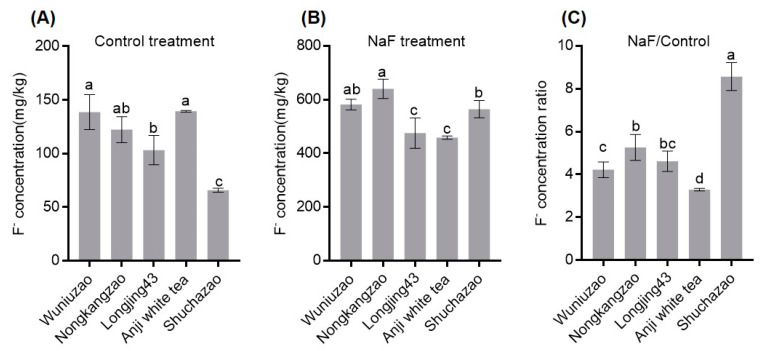
Effect of fluoride content in leaves of different tea varieties under F^−^ treatment. (**A**) The F^−^ concentration in leaves of different tea varieties under on control treatment; (**B**) the F^−^ concentration in leaves of different tea varieties under NaF treatments; (**C**) ratio of F^−^ concentration in leaves of different tea varieties under NaF treatment to that control treatment. Data are shown as mean ± SD (*n* = 4 plants). Different letters indicate significant differences between treatments (*p* < 0.05, one-way ANOVA).

**Figure 2 ijms-23-07756-f002:**
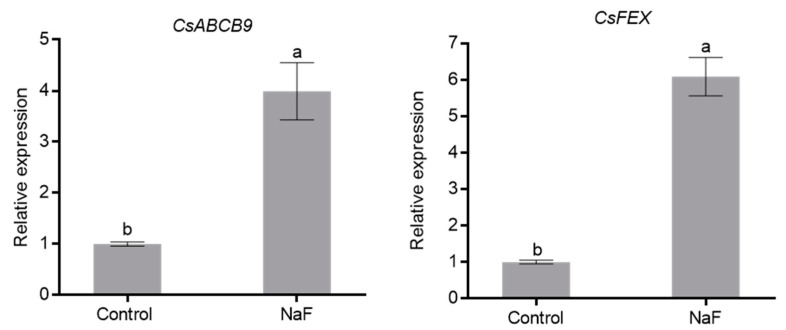
Expression patterns of *CsABCB9* and *CsFEX* in tea leaves under F^−^ treatment. Data are shown as mean ± SD (*n* = 4). Gene expression values under control condition were set as 1. Different letters indicate significant differences between treatments (*p* < 0.05, one-way ANOVA).

**Figure 3 ijms-23-07756-f003:**
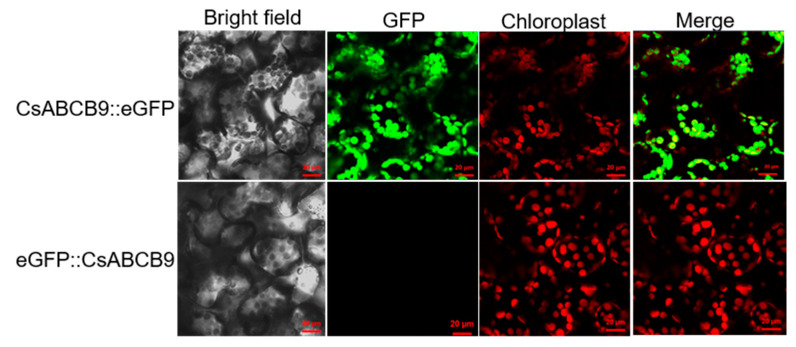
Subcellular location of CsABCB9. Epidermal leaf cells of N. benthamiana were infected with Agrobacterium GV3101 carrying GFP constructs 35S::eGFP, 35S::CsABCB9::eGFP and 35S::eGFP::CsABCB9. Bright-field, EGFP, chloroplast auto-fluorescence, and merged images were taken by confocal scanning microscopy, revealing that CsABCB9 was localized to the chloroplasts. Bars = 20 μm.

**Figure 4 ijms-23-07756-f004:**
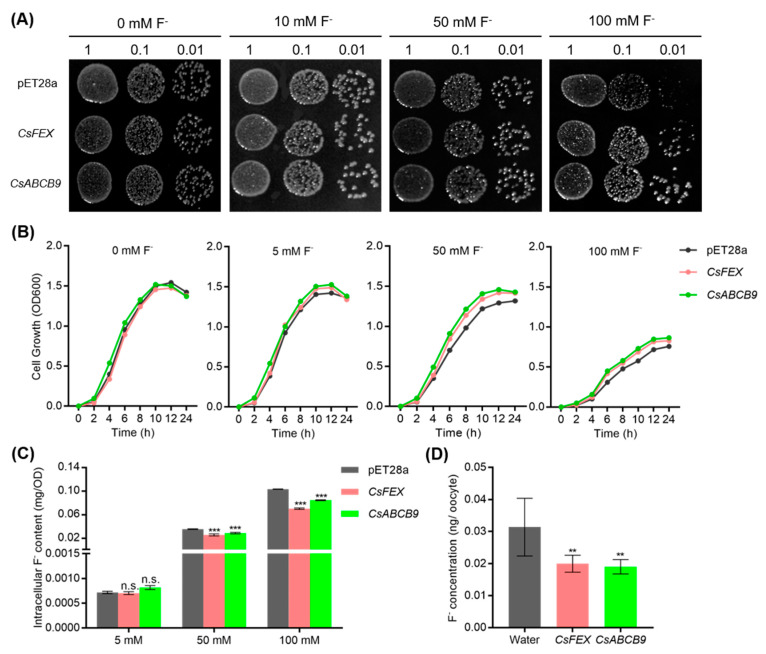
Functional analysis of *CsABCB9* in *Escherichia coli* and *Xenopus* oocytes. (**A**) Phenotypes of *Escherichia coli* colonies harboring empty vector (pET28a), *CsFEX* or *CsABCB9*. Washed bacteria solution of 10-fold serial gradient dilutions were spotted on culture plates containing different F^−^ concentrations (0, 5, 50, 100 mM). (**B**) Diagram of the growth curves of *Escherichia coli* cells transformed with pET28a, *CsFEX* or *CsABCB9* in different concentrations of F^−^ liquid medium. (**C**) Intracellular F^−^ content in *Escherichia coli* harboring pET28a, *CsFEX* or *CsABCB9* growing in different F^−^ concentration media. Data are shown as means ± SD (*n* = 4). n.s. indicates no significant difference compared with pET28a in 5 mM F group. Asterisks indicate significant differences (***, *p* < 0.001, one-way ANOVA) compared with the pET28a in 50 mM or 100 mM F^−^ concentration group. (**D**) F^−^ efflux activity of CsABCB9 in *Xenopus* oocytes. Oocytes were injected with water, *CsFEX* or *CsABCB9* mRNA and cultured in cultural solution containing 1mM F^−^ for 8 h. Water-injected or CsFEX-expressing oocytes were used as a negative or positive controls, respectively. Data are shown as means ± SD (*n* = 8). Asterisks indicate significant differences compared with water-injected oocytes (**, *p* < 0.01, one-way ANOVA).

**Figure 5 ijms-23-07756-f005:**
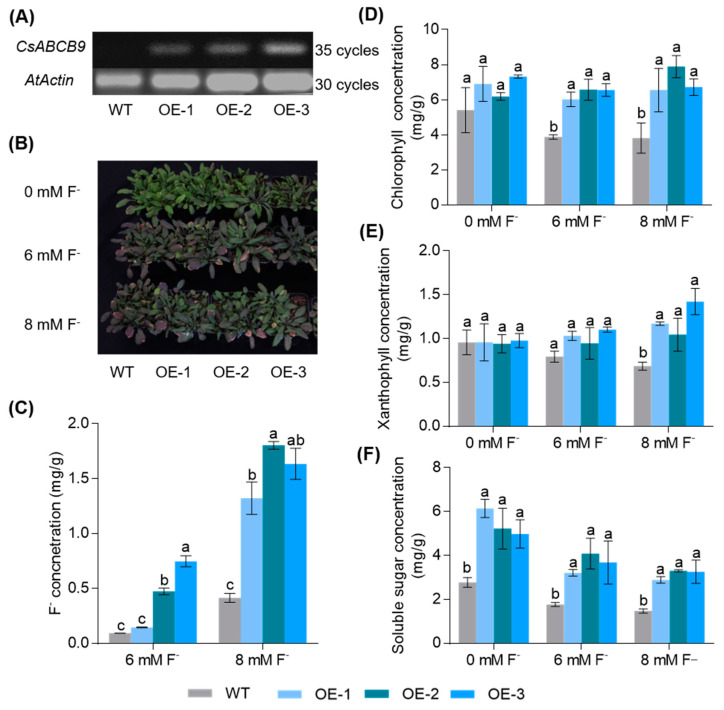
Effect of overexpression of CsABCB9 on fluoride resistance in *Arabidopsis*. (**A**) semi-quantitative quantification of *CsABCB9* in transgenic *Arabidopsis* using AtACTIN2 (AT3G18780) as an internal reference, (**B**) phenotypes of transgenic *Arabidopsis* and WT at different concentrations of F^−^ treatments, (**C**) F^−^ concentration in shoots of transgenic lines and WT *Arabidopsis* treated with different F^−^ treatments; (**D**) the concentration of chlorophyll, (**E**) xanthophyll and (**F**) soluble sugars in leaves of transgenic lines and WT *Arabidopsis* under different concentration of F^−^ treatments. Data are shown as means ± SD (*n* = 4 plants). Different letters indicate significant differences between overexpressed *Arabidopsis* lines and wild type (*p* < 0.05, one-way ANOVA).

**Figure 6 ijms-23-07756-f006:**
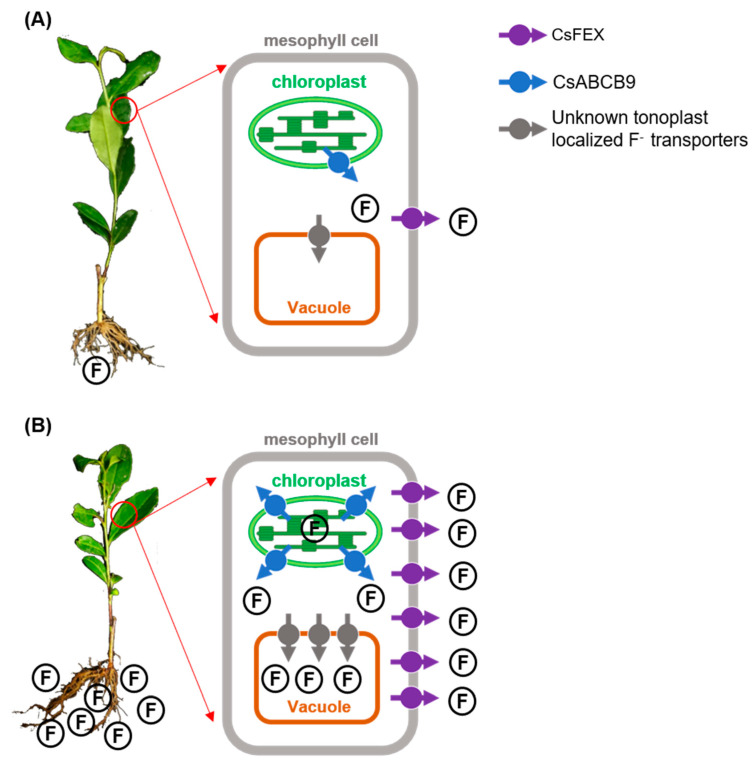
Working model of tea tree transporters involved in fluoride (F)-stress resistance. (**A**) Under no F or low F condition, the F transporters are only slightly expressed, and the tea plant can grow healthily. (**B**) Under F stress condition, F concentration in mesophyll cells is increased. Expression abundances of chloroplast-localized fluoride exporter CsABCB9 and plasmalemma-localized fluoride exporter CsFEX are increased. CsABCB9 and CsFEX transport F from the chloroplast to the cytosol and from the cytoplasmic matrix to the apoplast space, respectively. Moreover, some unknown tonoplast-localized transporters can leak F from the cytoplasm to the vacuole and protecting the organelle from damage caused by high F concentration in the cytoplasm.

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
