# Peer review of "Camellia sinensis* Chloroplast Fluoride Efflux Gene *CsABCB9* Is Involved in the Fluoride Tolerance Mechanism"

_ijms, 2022, doi:10.3390/ijms23147756_

Round 1

Author Response

Review 1

Comments and Suggestions for Authors

More or less text book knowledge tells us that the content in fluorides in soil can be considerable. However, usually only a small quantity is hydrosoluble, while the major part is fixed in minerals of the silicate and phosphate type. The normal concentration found is 0.3 ppm to 0.5 ppm, whereas the total content may vary from 10 to 1000 ppm.

In particular, in the proximity of aluminum foundries, of brickyards etc. dust particles and gas are emitted that contain fluorides which lead to concentrations in soil reaching several hundred kg per ha. But more importantly, in parallel the amount of hydrosoluble fluorides increases, at the extreme up to 25 ppm. Only in such soluble form plants can absorb fluorides, but it adds the absorption of gaseous HF via the stomata! Generally speaking, if the pH of soil is high and if there are high concentrations of Ca and phosphate, fluorine exists in form of insoluble CaF2 or Al2(SiF6)2. This situation changes when the soil becomes more acidic and the content in hydrosuluble fluorides increases, but even then, for instance observed with roots of Hordeum vulgare, the absorption of chlorides is about 100 times more efficient than that of fluorides. This explains the relatively low content of fluorine in plants. If higher, then it is frequently accompanied by a higher Al content. It was reported that this might lead to a drop in dry weight. Gaseous emissions of HF prorogue chlorosis in aerial parts, maybe to some degeneration of cytoplasm and structural changes in mitochondria, larger plastoglobules in chloroplasts and a loss in the lamellar structure of thylakoids (for instance in the moss Sphagnum nemeoreum).

It seems however that Camellia sinensis contains comparably high amounts of fluorine.

The manuscript here has examined how this plant can get rid of fluorine, involving a transporter localized to chloroplasts and another one localized to the plasma membrane. The combined action of both leads to export from cells potentially toxic (?) fluoride, as is summarized in some working model (Fig. 6). Through transcriptome analysis a gene (CsABCB9) was identified that encodes an ABC-type efflux transporter in chloroplasts. A functional proof was provided by heterologous expression in E. coli and Xenopus laevis oocytes, also in Arabidopsis thaliana, leading to resistance towards F-, in combination with a so-called FEX protein localized to the plasma membrane, encoded by CsFEX previously characterized by the same research group [ref. 17]. The study here is a logical continuation of what has been described in this reference 17. The experimental work looks solid and convincing. However, some criticisms are unavoidable. In the manuscript throughout the authors do not clearly distinguish between gene names (in ITALICS) and the proteins they code for! Keep in mind that genes don’t catalyze anything! It adds some weird writing in between, also some “wordy” style. Some condensation and regrouping of statements should increase the readability and understanding. Scientific names of species always in ITALICS throughout!

In the following some suggestions are made. It is recommended that the authors carefully check and consider newly written parts as suggested. A better and careful editing by the authors had made the life of the reviewer much easier! If there are questions and specific comments, then they will be marked with blue color.

Point 1: P 1, line 2: Already in the title write CsABCB9 in italics!

Respond 1: We have revised the format with tracker. Please see the line 2.

Point 2: Line 44: Camellia sinensis

Respond 2: We have revised the format with tracker. Please see the line 45.

Point 3: ff: ... found that a treatment with high concentrations of F reduced the photosynthetic rate and stomatal conductance [9, 10]. F can affect photosynthetic efficiency by preventing the dephosphorylation of thylakoid membrane proteins, especially the chlorophyll a/b protein complex [11]. In addition, F binds to magnesium in chlorophyll, causing thylakoids (to do what?) and thus inhibiting primary photosynthetic reactions. Moreover, F in leaves of tea plants can inhibit the activity of ribulose 1, 5-diphosphate carboxylase and thereby carbon assimilation [9, 12]. Therefore, the relative tolerance of tea plants to F may be closely related to its accumulation in the leaves. (This is a suggestion and hopefully corresponds to what the authors like to say.)

Respond 3: Thank you for your suggestions. We have revised these sentences. Please see the line from 160 to 168 in the revised version with tracker.

Point 4: P 2, line 59: ... in Saccharomyces cerevisiae membranes transport fluoride more specifically than chloride...

Line 61: ... ability to export fluoride ions [15]. FEX and Fluc are homologous genes (identified) in different species [14] and are involved in the detoxification of fluorine.

(Suggestion!)

Respond 4: Thank you for your suggestions. We have revised these sentences. Please see the line 173 and line 174 in the revised version with tracker.

Point 5: Line 63: ... FEX homologous genes...

Line 66 ff: CsFEX is primarily localized in the plasma membrane, and the expression of the corresponding gene CsFEX can be triggered by exogenous fluoride treatment [17]. However, how the fluorine efflux and accumulation as well operate at the molecular level remains still unclear, also which regulatory mechanisms are involved.

(Suggestion!)

Respond 5: We have revised these sentences according to your suggestion. Please see the line from 178 to 181 in the revised version with tracker.

Point 6: Line 70 ff: ... and hydrolyze it to generate energy to drive transport and regulate further cellular functions. ABC transporter family genes consist of eight subfamilies and the proteins coded for can transport many substrates. Thus they are involved in regulating (practically) all physiological and biochemical processes in plants [18].
Line 77: As a consequence, ABC transporters play an important role...
Line 78: Overexpression of AtABCG36 did improve desiccation and...
Line 79 ff: AtABCC5 and AtABCG31 (gene products) are involved in stress resistance in Arabidopsis by regulating the opening and closing of stomata in the leaf blade [22, 23]. In rice, the expression of the ABCB family genes is induced by by abiotic stress [24]. Based on those (and other) studies, it seems clear that ABC transporters regulate plant stress resistance during growth and development. These studies indicate that ABC transporters regulate stress resistance during plant growth and development. (Could be cut down. Essentially the same was already stated a few lines earlier!)

Line 85: ... functionally characterized from tea plants, although they are most possibly involved in the F- stress resistance mechanism in tea plants.

Respond 6: Thank you for your suggestions. We have revised these sentences. Please see the line from 182 to 197 in the revised version with tracker.

Point 7: Line 89: RNA-seq analysis led us to screen CsABCB9, a gene whose expression is increased by F treatment....
Line 94 ff: This enabled us to explore the function of CsABCB9 under F stress.
Consequently it was revealed that the CsABCB9 gene plays an essential role in F
resistance in tea and that in fact CsABCB9 is a fluoride efflux transporter gene that
enhances F resistance by protecting chloroplasts. (The following statement cut be cut completely!)

Respond 7: We have revised these sentences according to your suggestions. Please see the line from 201 to 208 in the revised version with tracker.

Point 8: P 3, line 99 ff: Leaves are the main source of fluorine enriched in tea. Therefore, we measured F- concentrations in leaves of different tea varieties treated with NaF for 3 days in hydroponic culture. Evidently there were significant differences in F concentrations in leaves from different tea varieties. Especially the F- accumulation in leaves of Wuniuzao was highest in comparison with that of other tea varieties (Figure1A and 1B). However, when the ratio between the F content of the F- treated tea varieties to the control condition was calculated, the highest value was seen for Shuchazao (Figure 1C). This indicates that Shuchazao is very sensitive to F treatment, possibly due to a strong F absorption capacity. Therefore, we selected Shuchazao as a representative tea species. The RNA of Shuchazao was extracted following the control versus NaF treatment and was analyzed by transcriptome sequencing. Here, the ABC transporter gene family contains 910 labeled genes, of which the ABC transporter B subfamily represents the highest number (Table S1).
Line 111 ff: Transcriptome sequencing, and the statistical analysis of differentially
expressed ABC transporter family genes revealed that the ABC transporter B subfamily had three genes with increased and 11 genes with decreased expression, for a total of 14 genes. In particular CL667 showed a 3.89-fold increase in expression (Table S2).
In addition, a preliminary analysis of PFMK (write out!) values in transcriptome
sequencing revealed that NaF treatment of tea leaves significantly increased the
expression of CL667 (Figure S1). This suggested that CL667 may be involved in the
regulation of F- absorption and transport in tea plants. At the gene prediction site
(http://tpdb.shengxin.ren/index.html) CL667 was also found to code for the ATP binding cassette, subfamily B member 9, and CL667 was thus named CsABCB9.
(Suggestion!)

Respond 8: Thank you for your suggestions. We have revised these sentences. Please see the line from 397 to 418 in the revised version with tracker.

Point 9: Line 130: We further examined the relative expression patterns of CsABC9 in leaves and roots under different F- treatments.
Line 132 ff: Clearly, the expression level of CsABCB9 in leaves under NaF stress was higher than in controls (Figure 2), well in agreement with the pattern from RNA-Seq (Figure S1). However, there were no significant differences in CsABCB9 expression in roots under the different treatments (Figure S2). Furthermore, the expression of CsFEX, previously reported as an F transport gene in tea, was induced by the external NaF (Figure 2 and Figure S2) [17].
P 4, line 145: According to the Gene Structure Display Server analysis, the total length of the CsABCB9 cDNA sequence is 2,128 bp, with an ORF sequence of 684 bp, encoding 227 amino acids (Figure S2A). The ORF sequence of CsABCB9 from tea leaves cDNA sequences was cloned (Figure S2B). (Transit peptide? Comparison with other genes coding for plastid-imported proteins?)
Line 150 ff: ... shows the calculated tertiary structure... (Consider to add literature
references!)
Line 152: SMART analysis (literature for this method?) indicated that CsABCB8 contains two conserved domains typical for the ABC transporter family, namely NBDs (nucleotide-binding domains) and TMDs (transmembrane domains) (Figure S3B).
Line 156 ff: The subcellular localization software Cell-PLoc 2.0 predicted that CsABCB9 is localized to chloroplasts for experimental confirmation, two vectors placed under the control of the CaMV 35S promoter were constructed expressing CsABCB9::GFP and GFP::CsABCB9, followed by infiltration of tobacco leaf epidermal cells. (Suggestion!)

Line 159 ff: (Any specific explanation for why the N-terminal GFP construct was not expressed at all?)

P 5 line 172: ... was a fluoride export gene [17].
Line 173: ... CsABCB9 overexpressing...
Line 176: ... exhibited a higher survival rate under...
Line 176 ff: E. coli strains overexpressing both CsABCB9 and CsFEX contained lower F contents than the negative control strain, a proof that CsABCB9 plays a role in enhancing resistance to F (Figure 4C). (This is a suggestion. But as it was written, it makes no sense! Don’t forget all the ITALICS in the legend to this figure!)

P 6 line 196: mediums media
Line 204: ... is still under investigation (?) ... under development (?)
Line 206 ff: Overexpression of CsABCB9 was confirmed in transgenic Arabidopsis (OE1, OE2, OE3), but not in the WT. (Figure 5A). Chlorophyll concentration, xanthophyll (how determined? Is not detailed in the Methods section!), soluble sugar, F accumulation, and growth phenotypes of wild-type and all transgenic lines were checked to determine whether OE of CsABCB9 in plant affects growth and increases F resistance (Figure 5).
(No loss in information...)

Respond 9: We have revised these sentences according to your suggestions. Please see the line from 427 to 778 in the revised version with tracker. Thank you for your suggestions. Besides, the reason why the C-terminal fusion GFP protein of CsABCB9 was not expressed may be the C-terminal of CsABCB9 protein folded into inside, resulting in the C-terminal fusion GFP florescent can’t observe fluorescence. The determination of chlorophyll, xanthophyll and soluble sugar amount had been added in Methods section. Please see the line from 1236 to 1245 in the revised version with tracker.

Point 10: P 7, line 228: ... were searched for by...
Line 229 ff: Previously it has been shown that ABC transporters are clearly involved in plant stress resistance [18]. Many plants exposed to F- exhibit phenotypes that indicated physiological toxicity [25, 26]. It is well known that tea is an F- resistant crop plant and that tea leaves possess a very strong F accumulation capacity [27-29]. Nonetheless, many studies have revealed that high concentrations of fluoride do affect the physiology and biochemistry of this F- resistant plant, including a reduced photosynthesis, disruption of cellular ultrastructures, and changes in the leaf antioxidant system [30, 31]. Previous work has demonstrated that FEX family transporters located in the cell membrane contribute to resistance against fluoride toxicity [14, 32].

P 8, line 238 ff: The web-based structure simulation of CsABCB9 revealed that a fragment contains a conserved ABC domain and a transmembrane domain (Figure S2 and S3). (Please, be aware of the fact that this are simulations, but no proof like an X-ray structure! Don’t make a big story out of this...) Furthermore, F treatment resulted in the highest expression of CsABCB9 in leaves compared to roots (Figure 2, Figure S2). These observations indicate that the expression of CsABCB9 in tea leaves is closely related to the F content and that CsABCB9 expression is indeed triggered induced by external F- [33].
Line 244 ff: To further clarify the function of CsABCB9, CsABCB9 was transferred to prokaryotic E. coli, followed by treatment with different concentrations of F-. Clearly, overexpression of CsABCB9 increased F resistance and decreased the F level (Figure 4A, 4B and 4C). This function was similar to the positive control CsFEX reported by Zhu et al., 2019 [17]. Moreover, the expression of CsABCB9 in eukaryotic Xenopus oocytes also reduced intracellular F- levels (Figure 4D). Additionally, CsABCB9 was present in chloroplasts (Figure 3). This suggests that CsABCB9 protects chloroplasts mainly by efflux of F-, thereby enhancing the fluorine resistance in tea plants.
Line 245 ff: Previously, it has been shown that overexpression of AtABCB25 improved the Pb/Cd resistance in Arabidopsis [34]. AtABCB27 and its homolog in barley were implicated in the prevention of Al toxicity [35]. To determine the mechanism by which CsABCB9 drives F resistance, overexpression lines of the model plant Arabidopsis were established (Figure 5A); CsABCB9 overexpression (Consider to replace “overexpression” at it’s first mentioning by “OE” in the entire manuscript thereafter!) lines were found to be more resistant to exogenous F treatment. Accordingly, chlorophyll, xanthophyll, and soluble sugar concentrations were increased in comparison to the wild type (Figure 5).In agreement with the observation of subcellular localization of CsABCB9 (Figure 3), OE of CsABCB9 apparently enhanced the F resistance by protecting leaf chloroplasts.
Line 263 ff: In conclusion, this is the first determination and cloning of the ABC
transporter family gene CsABCB9 for F-specific export in tea. The expression of CsABCB9 in tea was found to be leaf-specific and induced by external F- treatment. Furthermore, OE of CsABCB9 in E. coli and Xenopus oocytes suppressed intracellular F- accumulation and thus led to improve F- resistance. (The authors should consider to speak of F- (luoride) treatments and resistance throughout the whole manuscript when not precisely meaning “fluorine”, according to the experimental design, namely delivering fluoride in water/buffer solution, not for instance as HF gas or similar!) These results directly demonstrate that chloroplast-localized CsABCB9 acts as a fluoride efflux transporter to mitigate F- toxicity, and that CsABCB9 OE in Arabidopsis can protect leaf photosynthesis at comparably high concentrations of external F-.
Line 271 ff: Taken together, our previous and actual studies infer that only a few F
transporters are expressed in tea plants to maintain the healthy growth under no F or low F condition (Figure 6A). A high concentration of F- increased its level in mesophyll cells through transport of F- into the cells via passive absorption and active ion channels [36]. F- is subsequently transported to various intracellular organelles, increasing the expression of the genes encoding the chloroplast-localized fluoride efflux transporter CsABCB9 and the plasmalemma-localized CsFEX. CsABCB9 transported F- from the chloroplast to the cytosol, reducing F- toxicity to the chloroplast, and then CsFEX exports F- from the cytoplasmic matrix into cell wall or apoplastic space to avoid fluorine poisoning. Besides, excess F could also be transferred to and stored in vacuoles by an unknown tonoplast-localized transporters to alleviate intracellular F- toxicity (Figure 6B). However, the precise molecular control mechanisms of F accumulation, toxicity and detoxification, and resistance in tea are worth to be explored further.
(A conclusion should be short!)

Respond 10: We have revised these sentences. Please see the line from 780 to 839 in the revised version with tracker.

Reviewer 2 Report

The reviewer approves the quality of data and thinks that these should be made accessible to the community. The manuscript is, however, not well written and needs re-writing as well as thorough revision.

The following points might appear less important, but they are not. Inconsistencies in instances that can easily be done correctly, do not increase confidence of readers in validity and soundness of the results provided.

The reviewer recommends to improve the manuscript considerably prior to publication. The mentioned points are only examples. There is much more to be considered in the same line. All this is meant and indeed necessary to improve readability and impact of the manuscript. 

One formal thing in advance: Do the addresses apply to all four authors throughout?

Please re-consider the use of chemical names and their normal designation in speaking and also writing. If the authors use chemical  nomenclature at all in the running text, they have to be precise. It is not F or  fluorine that is taken up, but F- or better fluoride. Sentences of the type "... Arabidopsis were inhibited by F treatment." are not acceptable.  You certainly did not treat the plants with a hologen gas. In any case, if F is used somewhere in the text, it is certainly not an abbreviation. Writing e.g. "... fluorine (F) ..." belongs more into a chemical textbook for beginners than in a scientific communication. 

The authors must take care to use precise wording and correct context. An example from the abstract is: "... discovered by sequencing transcriptome analysis ..." It is certainly not possible to sequence an analysis and indeed neither a transcriptome, too.

Use as few abbreviations as possible. This renders a text more readable. Abstracts must never contain any abbreviation. These are part of databases and must be readable wirthout the possibility to look at the main text. The reviewer reconmmends even to avoid E. for Escherichia. The same applies to other genus names. The authors should take into account that not all readers are familiar with all organisms. 

The Keywords document inconsistency in using chemical terms. It is - correctly - fluoride efflux, but why do the authors use fluorine tolerance. Do they mean something different?

The Methods paragraph needs to be more precise. It is not sufficient, just to say "RNA was extracted ... with (maybe 'by' is more appropriate) different treatments". Please specify! This is an important point for those who perform similar axperiments and for those who want to evaluate the quality of RNA preparations. You can also not "construct" amplicons into a vector. Please use reasonable wording and provide details about the cloning procedure. Their is much more in this part that needs improvement.

Maybe, the reviewer missed it. Please check again if the new sequence is accessible in databases and provide the accession number. 

In addition, the source of important chemicals needs to be mentioned. It is also mandatory to provide citations of the originally published procedures, e.g. all those techniques and procedures that the authors have not developed themselves. 

The reviewer will be glad to recommend the manuscript for publication after very careful revision. 

Author Response

Response to Reviewer 2 Comments

Review 2

Comments and Suggestions for Authors

The reviewer approves the quality of data and thinks that these should be made accessible to the community. The manuscript is, however, not well written and needs re-writing as well as thorough revision.

The following points might appear less important, but they are not. Inconsistencies in instances that can easily be done correctly, do not increase confidence of readers in validity and soundness of the results provided.

The reviewer recommends to improve the manuscript considerably prior to publication. The mentioned points are only examples. There is much more to be considered in the same line. All this is meant and indeed necessary to improve readability and impact of the manuscript.

Respond: Thank you for your suggestion. We have made extensive changes to this manuscript. Please see the new revised revision.

Point 1: One formal thing in advance: Do the addresses apply to all four authors throughout?

Respond 1: The four authors are from the same laboratory. Therefore, the address apply to all authors.

Point 2: Please re-consider the use of chemical names and their normal designation in speaking and also writing. If the authors use chemical nomenclature at all in the running text, they have to be precise. It is not F or fluorine that is taken up, but F- or better fluoride. Sentences of the type "... Arabidopsis were inhibited by F treatment." are not acceptable.  You certainly did not treat the plants with a halogen gas. In any case, if F is used somewhere in the text, it is certainly not an abbreviation. Writing e.g. "... fluorine (F) ..." belongs more into a chemical textbook for beginners than in a scientific communication.

Respond 2: Thank you for your suggestions. I confused the two words-“fluorine” and “fluoride”. I have revised it as fluoride or F-in this manuscript. Please see the new revision.

Point 3: The authors must take care to use precise wording and correct context. An example from the abstract is: "... discovered by sequencing transcriptome analysis ..." It is certainly not possible to sequence an analysis and indeed neither a transcriptome, too.

Respond 3: Thank you for your suggestion. We have revised these sentences. Please see the new revision.

Point 4: Use as few abbreviations as possible. This renders a text more readable. Abstracts must never contain any abbreviation. These are part of databases and must be readable without the possibility to look at the main text. The reviewer recommends even to avoid E. for Escherichia. The same applies to other genus names. The authors should take into account that not all readers are familiar with all organisms.

Respond 4: Thank you for your suggestions. We have changed the abbreviation to the full name. Please see the new revision.

Point 5: The Keywords document inconsistency in using chemical terms. It is - correctly - fluoride efflux, but why do the authors use fluorine tolerance. Do they mean something different?

Respond 5: Thank you for your suggestion. I mistake the two words-“fluoride” and “fluorine”. Fluoride in this text is correct. “fluorine tolerance” was corrected to “fluoride tolerance”. Please see the Keywords part.

Point 6: The Methods paragraph needs to be more precise. It is not sufficient, just to say "RNA was extracted ... with (maybe 'by' is more appropriate) different treatments". Please specify! This is an important point for those who perform similar experiments and for those who want to evaluate the quality of RNA preparations. You can also not "construct" amplicons into a vector. Please use reasonable wording and provide details about the cloning procedure. There is much more in this part that needs improvement.

Respond 6: Ok, I have revised and supplemented the Method and Material part. Thank you for your suggestion.

Point 7: Maybe, the reviewer missed it. Please check again if the new sequence is accessible in databases and provide the accession number.

Respond 7: The accession number of CsABCB9 is TEA013868.1. You can find the CsABCB9 sequence from website (http://tpdb.shengxin.ren/index.html). I have added accession number in the manuscript. Please the line 418 in the revised version with tracker.

Point 8: In addition, the source of important chemicals needs to be mentioned. It is also mandatory to provide citations of the originally published procedures, e.g. all those techniques and procedures that the authors have not developed themselves.

Respond 8: Ok, I have added these information. Please see the Materials and Methods part in the revised version.

Point 9: The reviewer will be glad to recommend the manuscript for publication after very careful revision.

Respond 9: Thank you for your suggestions.

Reviewer 3 Report

1. Authors should revised cited web pages and provided updated links.

2. Many of the cited references have more than 10 years.

3. Authors should proved more details about the sample preparations and the experimental design applied for this research.

Author Response

Response to Reviewer 3 Comments

Review 3

Comments and Suggestions for Authors

Point1. Authors should revised cited web pages and provided updated links.

Respond 1: We have revised cited web pages in the new version and added relevant references. Thank you for your suggestion.

Point 2. Many of the cited references have more than 10 years.

Respond 2: Ok, we have deleted some older references.

Point 3. Authors should prove more details about the sample preparations and the experimental design applied for this research.

Respond 3: We have revised and supplemented the content of the Material and Methods. Please see the new revision. Thank you for your suggestion.

Round 2

Reviewer 1 Report

Good that essentially all suggestions have been followed...  The reference on chloroplast pigment determination is not very convincing and is certainly based on easily available literature with exact descriptions how to evaluate UV/VIS measurements. But this is not a crucial point now.

Reviewer 3 Report

Authors have considered all suggestions and comments provided to improved this manuscript.